# Forest Age Drives the Resource Utilization Indicators of Trees in Planted and Natural Forests in China

**DOI:** 10.3390/plants13060806

**Published:** 2024-03-12

**Authors:** Xing Zhang, Xiaohong Chen, Yuhui Ji, Ru Wang, Jie Gao

**Affiliations:** 1Key Laboratory for the Conservation and Regulation Biology of Species in Special Environments, College of Life Science, Xinjiang Normal University, Urumqi 830054, China; zxyybh@163.com (X.Z.); cheng1352023@163.com (X.C.); jjyh1757838695@163.com (Y.J.); 2Key Laboratory of Earth Surface Processes of Ministry of Education, College of Urban and Environmental Sciences, Peking University, Beijing 100863, China

**Keywords:** key leaf traits, resource utilization strategy, natural forests, planted forests, forest age, climatic factors, soil nutrient

## Abstract

Specific leaf area (SLA) and leaf dry matter content (LDMC) are key leaf functional traits commonly used to reflect tree resource utilization strategies and predict forest ecosystem responses to environmental changes. Previous research on tree resource utilization strategies (SLA and LDMC) primarily focused on the species level within limited spatial scales, making it crucial to quantify the spatial variability and driving factors of these strategies. Whether there are discrepancies in resource utilization strategies between trees in planted and natural forests, and the dominant factors and mechanisms influencing them, remain unclear. This study, based on field surveys and the literature from 2008 to 2020 covering 263 planted and 434 natural forests in China, using generalized additive models (GAMs) and structural equation models (SEMs), analyzes the spatial differences and dominant factors in tree resource utilization strategies between planted and natural forests. The results show that the SLA of planted forests is significantly higher than that of natural forests (*p* < 0.01), and LDMC is significantly lower (*p* < 0.0001), indicating a “faster investment–return” resource utilization strategy. As the mean annual high temperature (MAHT) and mean annual precipitation (MAP) steadily rise, trees have adapted their resource utilization strategies, transitioning from a “conservative” survival tactic to a “rapid investment–return” model. Compared to natural forests, planted forest trees exhibit stronger environmental plasticity and greater variability with forest age in their resource utilization strategies. Overall, forest age is the dominant factor influencing resource utilization strategies in both planted and natural forests, having a far greater direct impact than climatic factors (temperature, precipitation, and sunlight) and soil nutrient factors. Additionally, as forest age increases, both planted and natural forests show an increase in SLA and a decrease in LDMC, indicating a gradual shift towards more efficient resource utilization strategies.

## 1. Introduction

Key leaf traits refer to the morphological, physiological, and phenotypic characteristics of plants, which reflect their adaptability to the environment, survival strategies, and ecological niches [1]. Specific leaf area (SLA) and leaf dry matter content (LDMC) are commonly used to represent tree resource utilization strategies [1,2]. SLA refers to the ratio of leaf area to its dry weight, while LDMC refers to the ratio of leaf dry weight to its fresh weight. Trees with higher SLA and lower LDMC are characterized by higher photosynthetic efficiency, minimized costs for leaf construction and maintenance, and maximized capacity for resource acquisition, indicative of a “faster investment–return” resource utilization strategy [3]. Exploring the resource utilization strategies of forest trees holds significant value for the rational management and operation of forests.

Since natural forests are largely undisturbed by human activities, previous research on forests has mainly focused on natural forests. However, from 1990 to 2020, the global area of planted forests increased by 123 million hectares [4,5,6]. Therefore, the ecological functions of plantation forests, such as carbon sequestration, are gradually receiving attention. Compared to natural forests, planted forests typically have less species diversity, are more homogenous, and are largely subject to human management [6]. The differences, if any, in resource utilization strategies between trees in planted and natural forests, as well as the dominant factors and mechanisms due to different forest origins, remain unclear.

SLA and LDMC are frequently used to reflect trees’ resource utilization and allocation strategies [2,7]. In regions with abundant water and thermal conditions, trees reduce leaf construction costs and enhance photosynthetic capacity by increasing SLA and decreasing LDMC, thus adopting a fast investment–return resource utilization strategy [3,7]. Trees in harsh, resource-scarce environments tend to have lower SLA and higher LDMC to maximize their lifespan, employing a conservative resource utilization strategy—slower investment–return [7,8]. Consequently, environmental factors play a crucial role in shaping the spatial variability of resource utilization strategies across different types of trees [9].

Numerous studies have found that climatic factors, especially temperature and precipitation, significantly affect tree nutrient utilization strategies [6,9]. Higher temperatures and abundant rainfall largely alleviate environmental survival stress for trees, leading them to allocate more resources to compete for light. They increase specific leaf area (SLA) to enhance photosynthetic capacity and effectively utilize light resources [10,11]. Conversely, in cold and arid conditions, the survival stress of trees increases, forcing them to devote more resources to survival. They reduce SLA and increase LDMC to extend lifespan and slow down growth, tending towards a conservative resource utilization strategy [12]. Sunlight is also a key climatic factor affecting tree resource utilization strategies [13]. Extensive research indicates that under prolonged and intense sunlight, trees are compelled to reduce SLA and increase LDMC, thereby lowering their photosynthetic rate to mitigate the harm from solar radiation, leading to a more conservative resource utilization approach [14]. The differences in species composition and management between planted and natural forests imply significant variations in how different species respond to climate change in terms of resource utilization strategies. Investigating the key climatic factors that drive resource utilization strategies in planted and natural forests is of significant ecological importance.

Soil nutrient factors, such as nitrogen (N), phosphorus (P), and pH, indirectly influence tree resource utilization strategies by affecting the absorption of nitrogen and phosphorus and the availability of nutrients [15,16,17]. Trees with high SLA and low LDMC typically adopt a fast investment–return strategy to better adapt to nutrient-rich soil conditions. Under conditions of high nitrogen and phosphorus, trees often increase leaf area (higher SLA) and reduce the proportion of leaf dry weight to enhance photosynthetic efficiency, optimizing their growth and development, minimizing nutrient investment in support structures and durability, and investing more nitrogen and phosphorus to support leaf growth [11]. In environments with low nitrogen and phosphorus, trees conserve more nutrients in longer-lived and more resilient tissues, adopting a conservative resource utilization strategy to adapt to these conditions [18]. Acidic soils may fixate on certain nutrients, making them unavailable, which might force trees to adopt a rapid growth strategy to better utilize the limited available nutrients. Different types of trees also exhibit significant variability in their adaptability to soil pH [19]. Research has found that under acidic conditions, trees have higher SLA and lower LDMC, enabling them to acquire a large amount of nutrients in a short time for maximal growth [20].

In addition to abiotic factors such as climate and soil nutrients, stand factors like stand density and forest age also influence tree resource utilization strategies [21]. Trees adopt different resource utilization strategies at various stages of forest development (forest age) to cope with survival pressures such as intra- and interspecies competition [21,22,23]. In the early stages of forest development, when species are more homogenous and stand density is relatively low, the forest canopy is smaller, light resources are abundant, and trees grow rapidly. During this period, trees generally have higher SLA and lower LDMC, adopting a “fast investment–return” survival strategy [3,21]. In the later stages of forest development, when the canopy structure becomes more complex, trees tend to have lower SLA and higher LDMC, adopting a conservative resource utilization strategy to adapt to intense light competition [14].

This study, based on field surveys and the literature from 2008 to 2020 covering 263 planted forests and 434 natural forests in China, aims to explore the spatial differences and dominant factors in tree resource utilization strategies between planted and natural forests (Figure 1). To address these issues, we propose the following hypotheses: (1) the resource utilization strategies of trees in planted forests are more conservative compared to those in natural forests; (2) abiotic factors, such as climatic and soil nutrient factors, are the primary drivers of the large-scale spatial variability in tree resource utilization strategies in planted and natural forests, with stand factors (forest age, stand density) also playing a significant role; (3) the direct effects of climatic factors on the resource utilization strategies of trees in planted and natural forests are greater than their indirect effects (modifying soil nutrient availability and stand factors).

## 2. Results

The specific leaf area (SLA) of planted forests was significantly higher than that of natural forests (*p* < 0.01, Figure 2B), while the leaf dry matter content (LDMC) was significantly lower (*p* < 0.0001, Figure 2C). This pattern represents a “rapid investment–return” resource use strategy.

With changes in the average temperature of the hottest month (MAHT), both planted and natural forests showed similar trends in SLA and LDMC, indicating a convergent effect of their resource use strategies in response to temperature variations (Figure 3A,D). With the continuous increase in MAHT and mean annual precipitation (MAP), the trees’ resource use strategy shifted from a “conservative” survival strategy to a “rapid investment–return” approach. The SLA and LDMC of planted and natural forests exhibited opposite trends with changes in MAHT, MAP, and average sunshine duration (ASD) (Figure 3). MAHT had the strongest explanatory power for the spatial variability of SLA (*R*^2^ = 0.19, Figure 3A), while ASD had the strongest for LDMC (*R*^2^ = 0.11, Figure 3F).

Overall, the resource use strategies (SLA and LDMC) of natural forests were relatively stable with changes in soil nutrients, while those of planted forests were more variable (Figure 4). With increasing forest age, both planted and natural forests showed a gradual increase in SLA and a decrease in LDMC, making their resource use strategies more efficient (Figure 5).

The potential influencing factors of the forest trees’ resource use strategies (SLA and LDMC) were significantly correlated with each other (*p* < 0.05, Figure 6). Variance decomposition results indicated that the resource use strategies (SLA and LDMC) of planted forest trees had greater environmental plasticity (higher *R*^2^ than natural forests) (Figure 7). The resource use strategies (SLA and LDMC) of natural forests mainly depended on climatic and stand factors, while those of planted forests depended mainly on stand factors (Figure 8). Overall, forest age was the dominant factor determining the resource use strategies (SLA and LDMC) of both natural and planted forests (Figure 8). The structural equation model results showed that stand factors (age and density) had a much greater direct impact on the resource use strategies (SLA and LDMC) of natural and planted forests than climatic and soil factors (Figure 9).

## 3. Discussion

This study found that the SLA of planted forests is significantly higher than that of natural forests, while the LDMC is significantly lower (Figure 2). There are significant differences in the resource use strategies of trees in natural and planted forests [3,11]. Planted forests typically adopt a rapid investment–return resource use strategy, enabling trees to grow at a faster rate [3]. Compared to natural forests, the resource use strategy of planted forests is generally more aggressive, largely due to differences in their growth environments, ecological dynamics, and evolutionary backgrounds [6,24]. Compared to natural forests, planted forests have lower species diversity, a simpler structural composition, and are more influenced by human activities and management. They also have more nutrient sources, as planted forests often increase soil nutrients (soil nitrogen and phosphorus content) through fertilization to promote rapid tree growth [6].

With the continuous increase in temperature and precipitation, the SLA tends to increase, while the LDMC shows a decreasing trend. The resource use strategy of trees shifts from a “conservative” survival strategy to a “rapid investment–return” strategy [3]. Compared to natural forests, planted forests demonstrate greater plasticity in their resource use strategy in response to climatic changes. This is mainly because natural forests, with their higher species diversity, include tree species adapted to various climates, providing greater adaptability to environmental changes caused by climate change. In contrast, the monoculture composition of planted forests may make them more susceptible to climate change impacts [25,26]. Species in planted forests, selected for human timber needs, often have shorter growth cycles and higher growth rates, which may make them more sensitive to climate change. Therefore, the resource use strategies of trees in planted forests are usually more sensitive to environmental changes [27,28,29].

Similarly, compared to natural forests, the SLA and LDMC in planted forests exhibit more dramatic changes in response to variations in soil nitrogen, phosphorus, and pH. This is primarily due to the fact that nutrient sources in natural forests come from natural precipitation, microbial decomposition of plant residues, and the leaching of mineral elements, whereas soil nutrients in planted forests largely depend on fertilization and irrigation. The soil nutrient content in planted forests can be affected by planting methods and fertilizer use, leading to areas of nutrient enrichment or depletion [30]. Natural forests have a more complex ecological environment with a wide variety of plant species and rich structural layers, featuring more complex root systems. In contrast, planted forests have a relatively simpler ecological environment with fewer plant species and simpler structural layers, and their root systems are also relatively simple [31,32]. The complex root systems and diverse microbial compositions in natural forests contribute to a more intricate nutrient cycling system, providing higher stability in the face of changes in soil nutrients and pH. On the other hand, the monoculture composition and simple root structure of planted forests result in a more simplistic nutrient cycling system, making them more sensitive to changes in soil nutrients [33].

As forest age increases, the SLA of both planted and natural forests gradually increases, while the LDMC decreases. Trees transition from a “conservative” resource use strategy to a “rapid investment–return” strategy; the variability in SLA and LDMC with forest age is more pronounced in planted forests than in natural ones (Figure 5). With age, natural forests experience an increase in biodiversity, a more complex composition of tree roots and microorganisms, and a significant improvement in the efficiency of material cycling within the ecosystem [34,35]. As forest ecosystems evolve, microorganisms continually transform organic matter into inorganic matter by decomposing plant residues and dead organisms, gradually improving soil fertility, water retention, and nutrient-holding capacity [36,37]. Trees can obtain more nutrients from the soil for their growth. Moreover, with increasing forest age, the increased species diversity and complexity of tree root systems in natural forests lead to more stable responses to climate change [38]. In contrast, trees in planted forests grow quickly, have shorter life cycles, are composed of single species, and are more influenced by human activities. Their soil nitrogen and phosphorus inputs primarily rely on artificial fertilization [6]. Therefore, compared to natural forests, the resource use strategies in planted forests exhibit greater variability with changes in forest age.

The resource use strategy of natural forests is mainly influenced by climatic and stand factors, whereas that of planted forests is predominantly affected by stand factors (Figure 8). This is because climatic factors (such as temperature, precipitation, and sunlight duration) affect tree SLA and LDMC by influencing photosynthesis intensity, respiration rate, leaf transpiration, and the plant’s ability to absorb nutrients from the soil, thereby affecting the tree’s resource use strategy [39,40]. Stand factors (forest age and density), on the other hand, influence the forest’s resource use strategy by altering the species composition of the forest ecosystem, the response to environmental changes, and the availability of soil nutrients [4,15,41,42].

Stand factors (forest age and density) have a far greater direct impact on the resource use strategies (SLA and LDMC) of planted and natural forests than climatic and soil factors (Figure 9). Forest age and density directly influence the growth state of trees and competition for resources [43]. In planted forests, age and density are primarily regulated by human management, hence playing a more significant role in resource use strategies. For instance, young and densely planted forests may promote rapid tree growth, thus affecting SLA and LDMC. Conversely, in natural forests, these factors tend to be more diverse and less controlled by humans. With increasing age in natural forests, species diversity and interspecies competition intensify [29]. This is likely because stand density impacts the community’s use and allocation of environmental resources [44], and forest age significantly influences soil fertility and its organic carbon content [41,45]. Furthermore, the rate of change in stand factors (forest age and density) in planted forests is relatively rapid, reflecting the current state of tree growth and resource utilization [6]. Therefore, despite the important influence of climatic and soil factors on tree growth and resource utilization [46,47,48,49], the directness and responsiveness of stand factors (forest age and density) render their impact on the resource use strategies of natural and planted forests more pronounced.

Exploring the resource utilization strategies of planted and natural forest trees is of great importance for forest management and addressing global change. Firstly, by understanding the resource utilization strategies of different types of forests, managers can more effectively plan and implement forest management practices to enhance the growth efficiency of trees and the health of ecosystems [6,11,50]. For example, understanding the photosynthetic efficiency and water use efficiency of specific tree species under certain environmental conditions can aid in selecting tree species that are more suited to local environments for planting and restoration efforts [6]. Secondly, this knowledge is crucial for responding to global climate change. Different tree species respond differently to environmental factors such as temperature and precipitation, affecting their survival and reproduction under the backdrop of climate change [50,51,52,53]. Understanding these resource utilization strategies helps predict the potential impacts of climate change on forest ecosystems and develop more targeted adaptation and mitigation measures. Moreover, comparing the resource utilization strategies of plantation and natural forests can reveal the impact of human intervention on the functionality of forest ecosystems, thus providing a scientific basis for ecological restoration and sustainable forest management. This includes optimizing the structure of tree plantations, enhancing the carbon sequestration capacity of forests, and increasing the resilience and adaptability of ecosystems to environmental changes [30,52].

## 4. Materials and Methods

### 4.1. Research Area and Sample Data

China has abundant forest resources and a vast distribution of vegetation and has established more planted forests than any other nation [6,25]. To delineate the differences in resource utilization strategies between natural and planted forests, we analyzed data from 263 planted and 434 natural forests across China, gathered through field surveys and the literature reviews spanning 2008 to 2020. The detailed data sources are presented in Appendix A. In this study, the dataset encompassing 697 forests from 82 locations comprised two primary sources: 438 forest datasets from 51 sites were gathered through the literature review, while the remaining 259 forest datasets from 31 sites were obtained from experiments conducted in this research.

For each study site, we systematically selected a minimum of four adjacent forest plots (each 30 m × 30 m) representing the typical zonal vegetation. We precisely recorded the spatial coordinates for each site. To calculate community-weighted functional traits, we quantified the abundance (number of individuals) of each tree species within these plots. All trees were accurately identified by their scientific names, with species verifications conducted using herbarium specimens for rigor and accuracy.

### 4.2. Key Leaf Traits

SLA (specific leaf area) and LDMC (leaf dry matter content) are widely used to indicate plant resource utilization strategies [3,46,47]. In each designated forest plot, we meticulously collected leaves that were free from any pest or disease damage from more than 20 mature and fully grown trees per species. These samples were carefully harvested from the tree crowns’ outer branches using scissors, ensuring minimal disturbance to the tree structure. Subsequently, the leaves were sandwiched between two layers of damp filter paper and securely stored in zip-lock bags for transport to the lab. Upon arrival, these samples were promptly placed in a refrigeration unit. Before conducting further analysis, we removed the leaves from their bags and allowed them to incubate for 12 h in a dark environment at 5 °C to rid them of any superficial moisture. Finally, the leaves were delicately patted dry using filter paper to absorb any residual water before their saturated fresh weight was accurately measured on an electronic scale.

In our experiments, we utilized the leaf area meter to measure the area of freshly picked single leaves with petioles removed. The fresh weight of the leaves was determined using an electronic scale with a precision of 0.0001 g. Fresh leaves were then placed in an oven at 105 °C for 15 min to kill green tissue, followed by drying at 60 °C for 48 h, after which their dry weight was measured using an electronic scale accurate to one ten-thousandth. The specific leaf area (SLA) (m²/kg) is calculated as leaf area divided by leaf dry weight, and leaf dry matter content (LDMC) (g/g) as leaf dry weight divided by leaf fresh weight.

Due to the significant differences in species abundance, the arithmetic mean of functional traits does not adequately represent the overall level of community functional traits [48,49,50,51,52,53]. Therefore, we used a community-weighted mean trait (CWM) based on species abundance to represent the average functional traits of planted and natural forests.
(1)$$\mathrm{CWM}=\sum_{i=1}^{S}D_{i}\times Trait_{i}$$

Here, CWM represents the weighted characteristic value of community functional traits, D_i_ represents the abundance of each tree species, and Trait_i_ indicates the selected functional trait.

### 4.3. Climatic, Soil, and Stand Data

Data for the hottest month’s average temperature (MAHT) and mean annual precipitation (MAP) were extracted at a 1 km spatial resolution from WorldClim (https://www.worldclim.org, accessed on 1 January 2023). Sunlight is a key environmental factor indicating plant resource utilization strategies [13]. ASD (annual average sunshine duration) data were sourced from the Meteorological Data Center of the China Meteorological Administration (http://data.cma.cn/site/index.html, accessed on 1 January 2023). We extracted data on soil pH, total soil nitrogen (http://www.csdn.store, accessed on 31 October 2022), and available soil phosphorus (https://www.osgeo.cn/data/wc137, accessed on 31 January 2023) from the top 30 cm of soil in a grid with a resolution of 250 m. Stand factors include forest age and stand density. Forest age was determined by consulting the relevant literature and field inquiries, with stand density calculated as the number of individual trees per unit area of the plot.

### 4.4. Data Analysis

We employed *t*-tests at a 0.05 significance level to examine whether there are significant differences in SLA and LDMC between planted forests and natural forests. This analysis was conducted in the R language package “agricolae” (version 4.3.1, R Core Team, 2023).

Generalized Additive Models (GAMs) were used to explore the effects of various abiotic factors (MAHT, MAP, ASD, soil nitrogen and phosphorus, and pH) and stand factors (age, density) on the spatial variation in SLA and LDMC. R^2^ represents the goodness-of-fit of the model, and *p*-value indicates significance. GAMs can be summarized as
(2)$$g(E(Y_{i}))=\beta_{0}+S_{1}(x_{i})+S_{2}(x_{i})+e_{i}\;$$
where g is a link function, *E*(*Y*_i_) is the estimate for the responsible variable *Y*_i_, S_1_ is the smooth function of xi over different light treatments, S_2_ is the smooth function of x_i_ along investigation time, x_i_ (i = 1, 2, 3, …, 12) are the explanatory variables, and they are number of new rhizomes, new rhizome length, new rhizome diameter, etc. β_0_ is constant term and ei is the error term. This analysis was performed in the “agricolae” package (version 4.3.1, R Core Team, 2023). To test for collinearity among these potential influencing factors, a heatmap was used to display the correlations between them, conducted in the “linkET” package in R.

Variance partitioning analysis was utilized to examine the contributions of climatic factors, soil factors, and stand factors to the resource use strategies of planted forests and natural forests, as well as the positive and negative effects of each potential influencing factor on the resource use strategies (SLA and LDMC) of planted forests and natural forests. This analysis was completed in the R package “rdacca.hp”. The machine learning boosted regression tree analysis method was used to explore the independent explanatory power of each potential influencing factor on the spatial variability of resource use strategies in planted forests and natural forests, with significance difference tests at a 0.05 significance level, conducted in the “gbm” package in R.

A piecewise structural equation model (piecewiseSEM) was used to investigate the impact pathways of climatic factors, soil nutrient factors, and stand factors on the resource use strategies (SLA and LDMC) of natural forests and planted forests. All observed variables were initially categorized into composite variables and integrated into the structural equation models (SEMs). To assess the robustness of the relationships between key ecological factors and both SLA and LDMC, we applied piecewiseSEM to account for the random effects of sampling points, providing insights into both the “marginal” and “conditional” roles of environmental predictors. This comprehensive analysis was carried out using the “piecewiseSEM”, “nlme”, and “lme4” packages in R. The model’s fit was evaluated using Fisher’s C-test. Criteria for an acceptable model included a significance level of *p* < 0.05 and an optimal model fit (0 ≤ Fisher’s *C*/*df* ≤ 2 and 0.05 < *p* ≤ 1.00). Subsequently, the model underwent a systematic refinement process, utilizing a stepwise approach for modification and enhancement.

## 5. Conclusions

This study, based on data from 263 planted and 434 natural forests in China, analyzes the spatial differences and dominant factors in tree resource utilization strategies between planted and natural forests. The results show that the SLA of planted forests is significantly higher than that of natural forests, and LDMC is significantly lower, indicating a “faster investment–return” resource utilization strategy. Compared to natural forests, planted forest trees exhibit stronger environmental plasticity and greater variability with forest age in their resource utilization strategies. Overall, forest age is the dominant factor influencing resource utilization strategies in both planted and natural forests, having a far greater direct impact than climate and soil nutrient factors.

## Figures and Tables

**Figure 1 plants-13-00806-f001:**
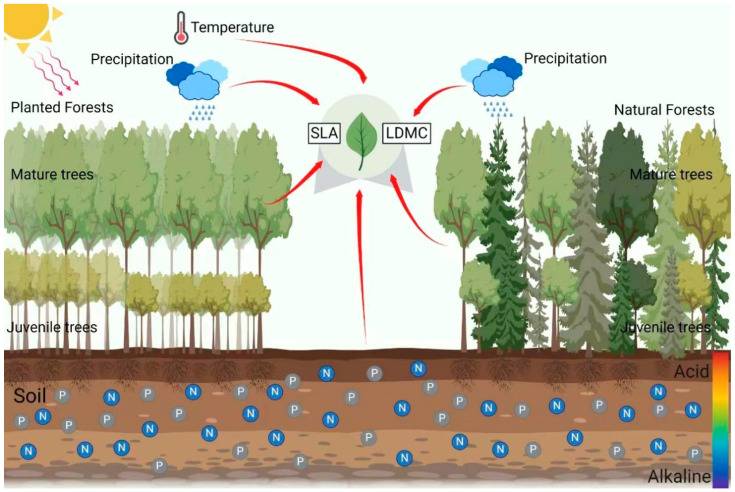
The schematic diagram for the study of the impact of climatic factors, soil factors, and stand factors on the specific leaf area (SLA) and leaf dry matter content (LDMC) in natural and planted forests.

**Figure 2 plants-13-00806-f002:**
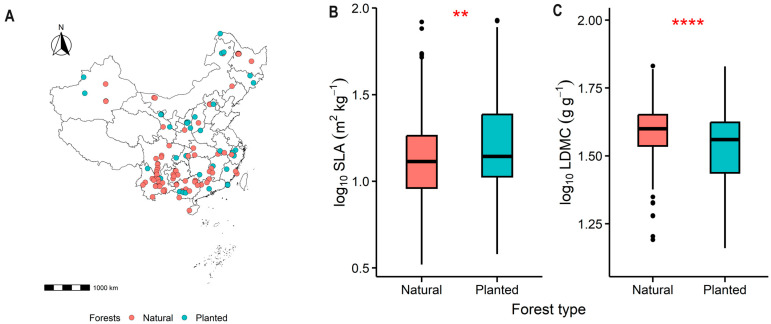
The geographical locations of the forest sample sites, (**A**), as well as the comparative differences in specific leaf area (SLA), (**B**), and leaf dry matter content (LDMC), (**C**), between natural and planted forests are presented. Both SLA and LDMC data were log-transformed. Asterisks indicate levels of significance (**** *P* < 0.0001; ** *P* < 0.01).

**Figure 3 plants-13-00806-f003:**
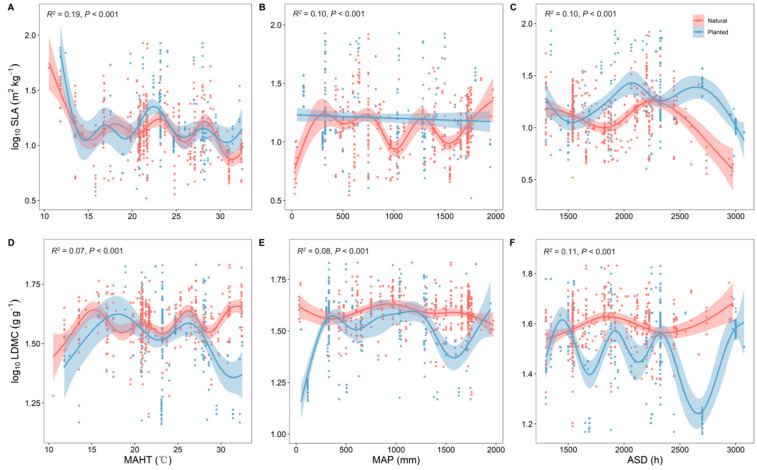
Correlation analysis of climatic factors with specific leaf area (SLA) and leaf dry matter content (LDMC) of natural and planted forests (**A**–**F**). Both SLA and LDMC data were log-transformed. MAHT represents the average temperature of the hottest month, MAP represents the mean annual precipitation, and ASD represents the annual average sunshine duration. *R*^2^ represents the goodness of fit for the Generalized Additive Model (GAM), and *P*-value indicates the level of significance.

**Figure 4 plants-13-00806-f004:**
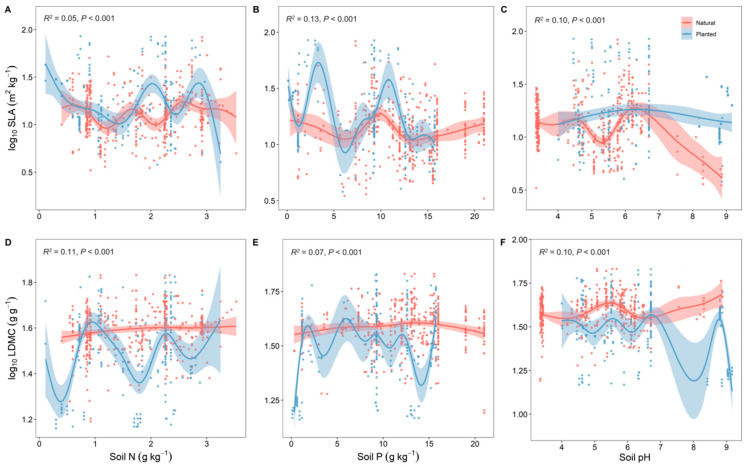
Correlation analysis of soil factors with specific leaf area (SLA) and leaf dry matter content (LDMC) of natural and planted forests (**A**–**F**). Both SLA and LDMC data were log-transformed. Soil factors include soil total nitrogen content (soil N); soil available phosphorus content (soil P); and soil pH. *R*^2^ represents the goodness of fit for the Generalized Additive Model (GAM), and *P*-value indicates the level of significance.

**Figure 5 plants-13-00806-f005:**
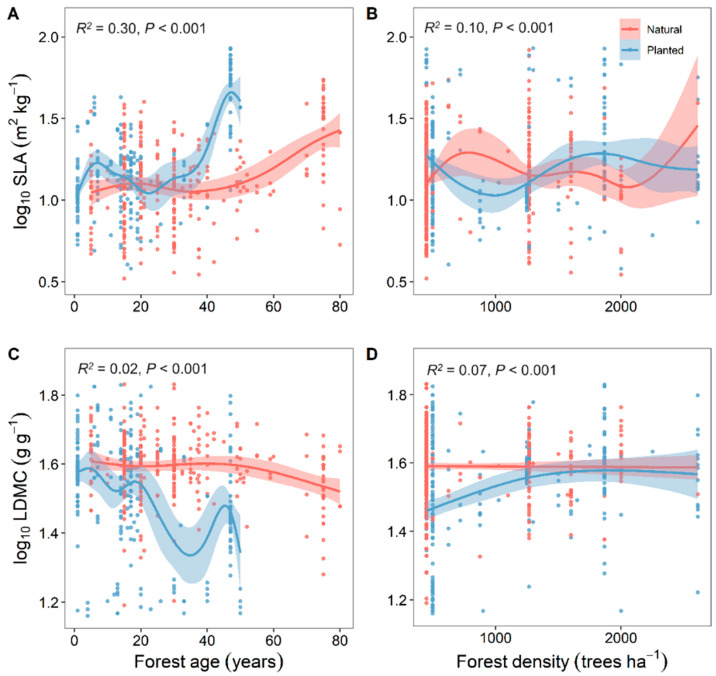
Correlation analysis of stand factors with specific leaf area (SLA) and leaf dry matter content (LDMC) of natural and planted forests (**A**–**D**). Both SLA and LDMC data were log-transformed. *R*^2^ represents the goodness of fit for the Generalized Additive Model (GAM), and *P*-value indicates the level of significance.

**Figure 6 plants-13-00806-f006:**
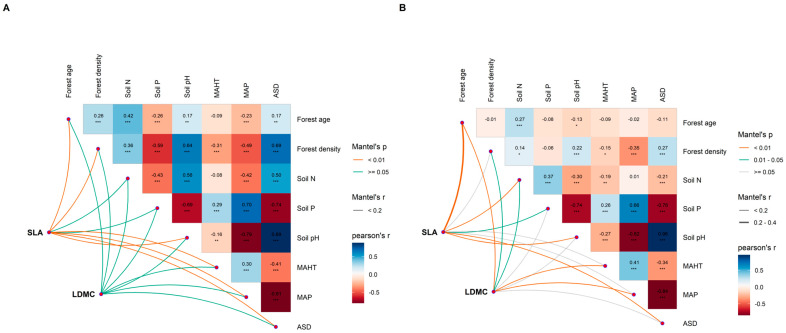
Multivariate correlation analysis of potential influencing factors of specific leaf area (SLA) and leaf dry matter content (LDMC) in natural forests (**A**) and planted forests (**B**). Both SLA and LDMC data were log-transformed. The influencing factors include climate factors (average temperature of the hottest month (MAHT), mean annual precipitation (MAP), and annual average sunshine duration (ASD)), soil factors (soil total nitrogen content (soil N), soil available phosphorus content (soil P), and soil pH), and stand factors (forest age and forest density). Asterisks indicate levels of significance (*** *P* < 0.001; ** *P* < 0.01; * *P* < 0.05).

**Figure 7 plants-13-00806-f007:**
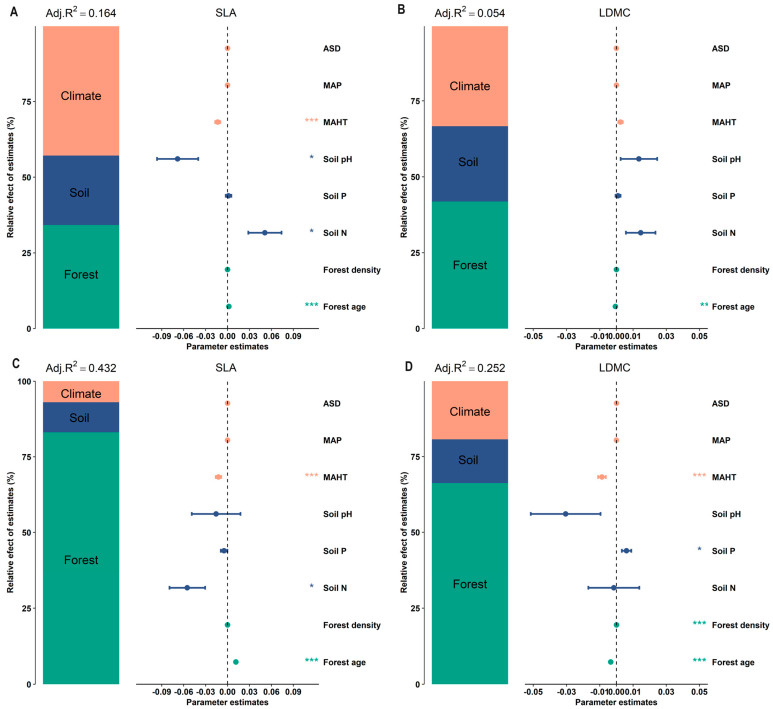
Relative effects of climate factors, soil factors, and stand factors on the specific leaf area (SLA) and leaf dry matter content (LDMC) in natural and planted forests. (**A**) SLA of natural forests. (**B**) LDMC of natural forests. (**C**) SLA of planted forests. (**D**) LDMC of planted forests. Both SLA and LDMC data were log-transformed. Climate factors include average temperature of the hottest month (MAHT), mean annual precipitation (MAP), and annual average sunshine duration (ASD). Soil factors include soil total nitrogen content (soil N), soil available phosphorus content (soil P), and soil pH. Stand factors include forest age and forest density. The averaged parameter estimates (standardized regression coefficients) of the model predictors are shown with their associated 95% confidence intervals along with the relative importance of each factor, expressed as the percentage of explained variance. The adjusted (adj.) *R*^2^ of the averaged model and the *p* value of each factor are given as *** *P* < 0.001; * *P* < 0.05.

**Figure 8 plants-13-00806-f008:**
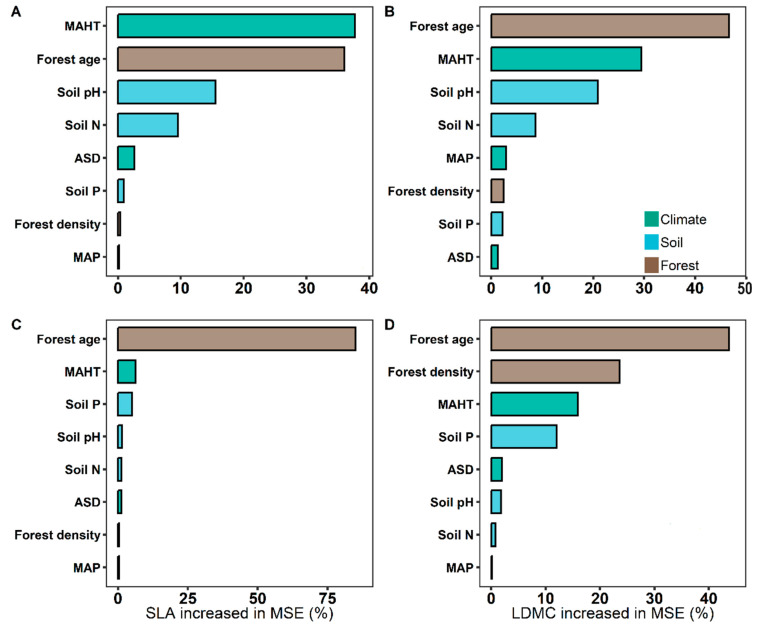
The relative importance of predictors in affecting specific leaf area (SLA) and leaf dry matter content (LDMC) of natural and planted forests. (**A**) SLA of natural forests. (**B**) LDMC of natural forests. (**C**) SLA of planted forests. (**D**) LDMC of planted forests. Both SLA and LDMC data were log-transformed. Climate factors include average temperature of the hottest month (MAHT), mean annual precipitation (MAP), and annual average sunshine duration (ASD). Soil factors include soil total nitrogen content (soil N), soil available phosphorus content (soil P), and soil pH. Stand factors include forest age and forest density. Percentage increase in mean square error (MSE, %) of variables was used to estimate the importance of these predictors, and higher MSE% values implied more important predictors.

**Figure 9 plants-13-00806-f009:**
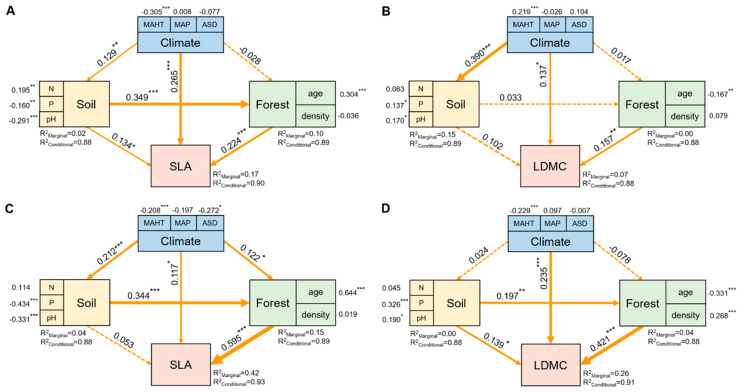
The direct and indirect effects of climate factors, soil factors, and stand factors on the specific leaf area (SLA) and leaf dry matter content (LDMC) of natural forests (**A**,**B**) and planted forests (**C**,**D**). Both SLA and LDMC data were log-transformed. Path diagrams represent the standardized results of final structural equation model (SEM) examining the relationships among variables. Numbers alongside the pathways indicate the standardized SEM coefficients, with asterisks indicating significant differences (*** *P* < 0.001; ** *P* < 0.01; * *P* < 0.05). The thickness of the arrow indicates the relative size of the path coefficient. R^2^ represents the proportion of variance for each explanatory variable.

## Data Availability

Data can be provided with request.

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
