# Peer review of "Forest Age Drives the Resource Utilization Indicators of Trees in Planted and Natural Forests in China"

_plants, 2024, doi:10.3390/plants13060806_

Round 1
Reviewer 1 Report
Comments and Suggestions for Authors
I did not find the study very interesting, as we can find many parameters that will differ between natural forests and plantations, without having further impact on management. However, I have no formal objections, and therefore I will leave it to the editors to decide whether such a study is in the intention of the journal.
Author Response
I did not find the study very interesting, as we can find many parameters that will differ between natural forests and plantations, without having further impact on management. However, I have no formal objections, and therefore I will leave it to the editors to decide whether such a study is in the intention of the journal.
Response:Thank you very much for your advice. Comparing the functional traits related to resource utilization strategy of planted and natural forests holds significant value for forest management. In 2023, we published our findings on the critical role of functional traits in predicting the functions of planted and natural forests in the international journals Journal of Integrative Plant Biology and Forest Ecology and Management. Of course, we highly respect your evaluation and hope to gain your approval.
Reviewer 2 Report
Comments and Suggestions for Authors
The manuscript, titled “Climatic Factors Determine the Resource Utilization Strategies 2 of Trees in Planted and Natural Forests at Macro Scales,” presents a study examining the influence of meteorological factors and soil on leaf size and dry matter content in natural and planted forests in China. The authors determined that planted forest trees exhibit higher ecological plasticity and greater variability depending on forest age in their resource use strategies than natural ones. The research is of interest for clarifying a number of environmental issues.
However, there are a number of questions and comments regarding this study:
- The final goal of this study is not entirely clear;
- Which strategy do the authors propose to use for natural forests, and which for planted forests?
- Have coniferous forests (natural and artificial) been studied and what dependencies were identified there?
- Was the height of forest growth and its influence on the factors studied taken into account?
- Was there any influence of zoning?
Author Response
The manuscript, titled “Climatic Factors Determine the Resource Utilization Strategies 2 of Trees in Planted and Natural Forests at Macro Scales,” presents a study examining the influence of meteorological factors and soil on leaf size and dry matter content in natural and planted forests in China. The authors determined that planted forest trees exhibit higher ecological plasticity and greater variability depending on forest age in their resource use strategies than natural ones. The research is of interest for clarifying a number of environmental issues. However, there are a number of questions and comments regarding this study:
Response: Thank you for recognizing our work. We have responded to your comments one by one based on your suggestions.
- The final goal of this study is not entirely clear;
- Which strategy do the authors propose to use for natural forests, and which for planted forests?
- Have coniferous forests (natural and artificial) been studied and what dependencies were identified there?
- Was the height of forest growth and its influence on the factors studied taken into account?
- Was there any influence of zoning?
Response: Our study investigates the impact of climatic factors, soil factors, and stand factors on the nutrient utilization strategies of different forest types by comparing the Specific Leaf Area (SLA) and Leaf Dry Matter Content (LDMC) of planted and natural forests. This provides a theoretical basis for understanding the effects of global change on forest management. Both resource utilization strategies identified in this study are applicable to planted and natural forests. However, the nutrient utilization strategies of these two types of forests may change in response to variations in climatic factors, soil nutrients, and stand factors. Coniferous forests exist in both planted and natural forests; however, this study focused solely on planted and natural forests. Thank you very much for your feedback; it will be a key focus of our future research. Generally speaking, forest height is closely related to forest age and stand density. Older forests typically have taller trees. Zoning may have an influence, but different zones have unique climate and soil conditions. Therefore, when exploring the specific impacts of climate and soil factors, we also quantified the contribution of zoning to a certain extent.
Reviewer 3 Report
Comments and Suggestions for Authors
General comments:
While the manuscript examines a topic that should be of considerable interest, it is need of significant improvement. This includes:
Moving the methods to before the results so that the reader can understand how the analysis was conducted.
More details on methods in terms of where leaf samples were collected, how they were collected and treated, how literature values were integrated with field data.
The findings are presented as definitive and causal, but are far from that. Correlational analyses were conducted and those cannot prove a causal relationship. They can suggest them, but that is not a definitive result. Also some of the correlations are extremely weak, yet are presented as if they all have biological significance. The authors need to be far more careful about how their results are presented and qualified.
It is never clear what a macro scale is. It seems to be a key justification for the study, yet it is never really defined or the issues with micro scale analyses being solved made clear. From what I can tell they did in fact do some analysis at the micro scale and so it is very confusing as to what they are learning really is.
There are contradictions in the manuscript that do not seem to make any sense. Some of this may be related to an imprecision of terminology. For example, the authors refer to plasticity, but what they are really discussing is responsiveness.
Specific comments (line):
2 Given this was a correlational study I do not think the word “determine” is appropriate. That implies a cause and effect relationship that cannot be determined using correlation. That is an overstatement of what the methods can tell one.
3 The location of the study should be made clear in the title.
13 limited is too vague. If the issue is the small spatial extent of prior studies then please write this. Or given the use of macro scale, perhaps micro scale would make even more sense.
14 “Due to differences in forest origins” is not needed as this is covered by latter part of the sentence.
18 There is nothing about the methods of how the analysis was done. These should be mentioned so that the reader understands the results summary.
22 These seems to be no uncertainty in the results. I doubt this was the case. I suggest the authors qualify these statements to be less definitive.
24-25 These statements seem to contradict the title. Which is correct?
28-30 This concluding sentence does not tell the reader anything new. It also employs circular logic. The study was done on X , so now we know more about X. Are there specific implications of the findings?
35 Here nutrients are mentioned, but on line 11 it was resources. Which is it? Please be consistent. Also please define the terms themselves so that the reader can understand the results and their implications more easily.
41 This sentence starts a completely new topic. It should start a new paragraph.
42 Is the point that the planted forests are more directly impacted? Both are impacted by human management.
44 What is the macroscale? Please define. Also it is not clear why this would be of interest.
35-45 I believe the problem is that this section does not do a very good job at introducing the manuscript. It seems to generally cover ideas that are more fully covered below. Perhaps this paragraph should be about the importance of understanding tree nutrient/resource strategies?
59 increase specific leaf area or leaf area?
68-69 Some of the logic behind this conclusion seems to be missing. Is this true because species have different strategies? Or because management creates different resource availabilities?
73 Don’t parts of the tree live outside the soil? This seems to suggest otherwise.
92 factors or attributes? The latter makes more sense.
96 please define stand density. Do the authors mean the number of trees per area or something else here? Also I find it hard to believe that the light resource is more abundant. Isn’t that determined by the length of day, the day of the year, the latitude, and the weather? How does stand density influence any of that? Do the authors mean that more of the light resource is available?
99 Is it solely a function of complexity? Isn’t largely due to the fact the canopy is more developed?
105 unfortunately figure 1 does not define the macro scale. It is actually a depiction at the micro scale or stand level. It would be extremely helpful for the authors to define what the macro scale is and to explain why one needs to study a phenomenon at this scale at all. I can imagine reasons, but the authors should have specific reasons in mind. Also it is not clear what the issues are. Finally, I don’t find these very compelling hypotheses. They just seem to justify the correlations that were examined, but don’t really examine any mechanisms.
115 I find it interesting that the trees do not seem to have any roots in the figure. I am not sure this figure does much more than just list factors, although the structural difference between planted and natural forests seems clearly illustrated.
119 Are there no methods? How is the reader supposed to understand the results if they have not been told the methods of analysis. Where did the data come from? How was it compiled or treated? How was it analyzed? Okay, I see that this information starts on line 298. It should go before the results.
122-123 This sentence does not make sense. The previous sentence compared two things. So is it the comparison that indicates a “rapid investment return strategy” or is it one of the two items being compared and which one?
140 Is correlation needed? It really wasn’t a correlation analysis per se. I also do not understand how such a complex curve was determined from one variable. Why is there a periodicity to the model. What mechanisms supports this complexity in the models?
216 Is it the forests that adapt or is it the trees?
223 This suggests that the planted forest fertilize themselves. Is that really the case?
227 It is not clear if these shifts are over time or over space. Can the authors be more specific?
229 I am not sure I am following the logic here. Natural forests are more diverse and include more species with differing traits which allows them to respond to a wider range of climatic conditions. But somehow this makes them less plastic than planted forests? There seems to be a disconnect. Is the issue that planted forests are more responsive to climatic changes? That is not clear from the text. Is more plastic the same as more sensitive?
281 Not according the title of the manuscript.
379 shouldn’t this be in past tense?
384 I don’t think plasticity is the issue here; isn’t it responsiveness?
387 not according to the title.
387-389 Again this statement does not tell the reader anything new.
300 I do not understand how forest coverage explains diversity or areal extent. This is not logical.
314 What were these measurements applied to exactly? To leave collected from where and how?
330 how was abundance determined? The relative basal area? The number of stems? The mass of the leaves?
345 I am not sure what this means. What were the specific significance tests?
Comments on the Quality of English LanguageThe main problem is odd word choices that do not have the meanings the authors are implying. See comments to authors for more details.
Author Response
Thank you very much for the valuable comments you provided on this article. I am the column editor for the journal Plants, where I have established a column on plant responses to climate change and contributed this article. My main research areas are forestry and ecology. I have carefully revised the article in response to your comments, hoping for mutual learning. My specific responses to your review comments are as follows:
General comments:
While the manuscript examines a topic that should be of considerable interest, it is need of significant improvement. This includes:Moving the methods to before the results so that the reader can understand how the analysis was conducted.
Response: Thank you for your suggestion; however, this is the format required by MDPI, and I hope you can understand.
More details on methods in terms of where leaf samples were collected, how they were collected and treated, how literature values were integrated with field data.
Response: SLA (Specific Leaf Area) and LDMC (Leaf Dry Matter Content) are widely used to indicate plant resource utilization strategies [3,46,47]. In each designated forest plot, we meticulously collected leaves that were free from any pest or disease damage from more than 20 mature and fully grown trees per species. These samples were carefully harvested from the tree crowns' outer branches using scissors, ensuring minimal disturbance to the tree structure. Subsequently, the leaves were sandwiched between two layers of damp filter paper and securely stored in zip-lock bags for transport to the lab. Upon arrival, these samples were promptly placed in a refrigeration unit. Before conducting further analysis, we removed the leaves from their bags and allowed them to incubate for 12 hours in a dark environment at 5℃ to rid them of any superficial moisture. Finally, the leaves were delicately patted dry using filter paper to absorb any residual water before their saturated fresh weight was accurately measured on an electronic scale.
In our experiments, we utilized the Japanese Cano Scan LIDE 110 portable leaf area meter to measure the area of freshly picked single leaves with petioles removed. The fresh weight of the leaves was determined using an electronic scale with a precision of 0.0001g. Fresh leaves were then placed in an oven at 105°C for 15 minutes to kill green tissue, followed by drying at 60°C for 48 hours, after which their dry weight was measured using an electronic scale accurate to one ten-thousandth. The Specific Leaf Area SLA (m² / kg) is calculated as leaf area divided by leaf dry weight, and Leaf Dry Matter Content LDMC (g / g) as leaf dry weight divided by leaf fresh weight.
In this study, the dataset encompassing 697 forests from 82 locations comprised two primary sources: 438 forest datasets from 51 sites were gathered through literature review, while the remaining 259 forest datasets from 31 sites were obtained from experiments conducted in this research. The data extracted from the literature were processed and calculated according to consistent experimental methods.
The findings are presented as definitive and causal, but are far from that. Correlational analyses were conducted and those cannot prove a causal relationship. They can suggest them, but that is not a definitive result. Also some of the correlations are extremely weak, yet are presented as if they all have biological significance. The authors need to be far more careful about how their results are presented and qualified.
Response: Thank you very much for your advice. In ecology or botany, there is never a direct cause-and-effect relationship. All relationships are presented based on correlation coefficients or the goodness of fit of models. We fully understand your concerns, but I believe we can discuss some practical issues through correlation coefficients, R2, or p-values. Frankly speaking, we have published several papers in CNS sub-journals (Nature climate change or Current biology), where, in macro studies, R2 might not exceed 0.1, but it still holds statistical value.Additionally, we employed structural equation modeling, a model widely used for the indirect relationships among multivariate. In any case, we are very grateful for your advice and will pay extra attention to this aspect in our subsequent research.
It is never clear what a macro scale is. It seems to be a key justification for the study, yet it is never really defined or the issues with micro scale analyses being solved made clear. From what I can tell they did in fact do some analysis at the micro scale and so it is very confusing as to what they are learning really is.
Response: Thanks for your views. We have clearly presented our research scale in Figure 2. The latitude range of China is 49 degrees, and the longitude range is 62 degrees. Therefore, if this scale does not qualify as a macro scale, it's hard for me to imagine that only a global study would. Whether it's considered a macro scale is not the focus of our research; we simply want readers to understand the biological patterns presented at such a broad geographical scale. Of course, to respect your opinion, we have removed the concept of "macro scale" from the title of our paper, hoping you can understand.
There are contradictions in the manuscript that do not seem to make any sense. Some of this may be related to an imprecision of terminology. For example, the authors refer to plasticity, but what they are really discussing is responsiveness.
Response: Plasticity refers to the degree of variation in the dependent variable as environmental factors change. Generally, we represent plasticity with the model's R2. This approach is widely used in the journal of Science or Nature.
2 Given this was a correlational study I do not think the word “determine” is appropriate. That implies a cause and effect relationship that cannot be determined using correlation. That is an overstatement of what the methods can tell one.
Response: I agree with your perspective. Therefore, we have changed "determine" to "drive." We hope this phrasing better aligns with your thoughts.
3 The location of the study should be made clear in the title.
Response: Thanks for your suggestions. We have added the location in the title.
14 “Due to differences in forest origins” is not needed as this is covered by latter part of the sentence.
Response: Thanks for your suggestions. We have excel this statement.
18 There is nothing about the methods of how the analysis was done. These should be mentioned so that the reader understands the results summary.
Response:This study, based on field surveys and literature from 2008 to 2020 covering 263 planted and 434 natural forests in China, using generalized additive models (GAMs) and structural equation models (SEMs) analyze the spatial differences and dominant factors in tree resource utilization strategies between planted and natural forests.
22 These seems to be no uncertainty in the results. I doubt this was the case. I suggest the authors qualify these statements to be less definitive.
Response:Thank you for your suggestion. Our statements are based on our research findings, and our results do not contain absolute terms (such as "definitely," "completely," etc.).
24-25 These statements seem to contradict the title. Which is correct?
Response: We have changed our title as “Forest Age Drives the Resource Utilization Indicators of Trees in Planted and Natural Forests in China”.
28-30 This concluding sentence does not tell the reader anything new. It also employs circular logic. The study was done on X , so now we know more about X. Are there specific implications of the findings?
Response: We have changed the statement: These findings have significant ecological implications for forest management.
35 Here nutrients are mentioned, but on line 11 it was resources. Which is it? Please be consistent. Also please define the terms themselves so that the reader can understand the results and their implications more easily.
Response: Thanks for your suggestions. We have changed the statement. Key leaf traits directly reflect tree resource utilization strategies.
41 This sentence starts a completely new topic. It should start a new paragraph.
Response: We agree with your opinions. This sentence starts a new paragraph.
42 Is the point that the planted forests are more directly impacted? Both are impacted by human management.
Response: The reasons why planted forests are more susceptible to impacts are: (1) Planted forests are usually composed of a single or a few tree species, lacking the biodiversity found in natural forests. This monoculture makes planted forests more vulnerable to specific pests and diseases due to the absence of natural control mechanisms and inter-species support. (2) Natural forests have developed a relatively stable ecological balance through long periods of evolution, whereas planted forests, with their shorter history of plantation and simpler ecosystems, lack sufficient time and conditions to form a stable ecological equilibrium, making them more susceptible to external disturbances. (3) The choice of tree species and growth conditions in planted forests are often based on economic or specific utility considerations rather than natural selection within ecosystems. This may lead to trees that are not adapted to local environmental conditions, thereby making them more susceptible to drought, diseases, and other impacts.
44 What is the macro scale? Please define. Also it is not clear why this would be of interest.
Response: In plant ecology, a very popular topic is the macro-scale changes characteristic of plant ecology. Common scales of study include macro scale, regional scale, and individual scale. This study is based in China, and therefore, it is on a macro scale.
35-45 I believe the problem is that this section does not do a very good job at introducing the manuscript. It seems to generally cover ideas that are more fully covered below. Perhaps this paragraph should be about the importance of understanding tree nutrient/resource strategies?
Response: Thank you for your feedback. This section is crucial for informing readers about the importance of tree survival strategies in responding to global changes. This is very important for those interested in studying plant ecology at a macro scale.
59 increase specific leaf area or leaf area?
Response: Thanks for your suggestions. That should be :increase specific leaf area.
68-69 Some of the logic behind this conclusion seems to be missing. Is this true because species have different strategies? Or because management creates different resource availabilities?
Response: Thanks for your advice. The differences in species composition and management between planted and natural forests imply significant variations in how different species respond to climate change in terms of resource utilization strategies.
73 Don’t parts of the tree live outside the soil? This seems to suggest otherwise.
Response: Thank you very much for your suggestion. From the perspective of plant ecology, it is true that parts of plants live in the air. However, plants primarily rely on the soil for their nutrient and water uptake. We have added the word "mainly" to clarify this point.
96 please define stand density. Do the authors mean the number of trees per area or something else here? Also I find it hard to believe that the light resource is more abundant. Isn’t that determined by the length of day, the day of the year, the latitude, and the weather? How does stand density influence any of that? Do the authors mean that more of the light resource is available?
Response: Stand density typically refers to the number of trees within a given area, which is one of the key indicators for measuring the structure of a forest or stand. Indeed, the total amount of light resources is determined by factors such as the length of day, the date within the year, latitude, and weather conditions, which are not affected by stand density. However, stand density affects the distribution and efficiency of light resource utilization beneath the canopy. In denser stands, competition among trees, especially for light, is more intense, which may result in reduced light reaching the understory and forest floor. Therefore, when mentioning that light resources are "more abundant," it could mean that in stands with lower density, the ground layer and understory plants have the opportunity to receive more light, thereby increasing the overall light utilization efficiency of the stand.
99 Is it solely a function of complexity? Isn’t largely due to the fact the canopy is more developed?
Response: In the later stages of stand development, the complexity of forest functions is not solely due to the development of the canopy. Although the development of the canopy directly impacts the forest's microclimate, light distribution, water cycle, and the growth of understory plants, thereby determining the ecological functions of the forest to some extent, the complexity of forest functions is influenced by a variety of factors. Different tree species and types of plants have varying impacts on ecosystem functions, and interactions between species also add to the system's complexity. The type, structure, and nutrient content of the soil have significant effects on the forest's water retention capability and nutrient cycling. Besides the canopy layer, the vertical and horizontal structure of the stand, including the understory vegetation and litter layer, also impacts the ecological functions of the forest.
105 unfortunately figure 1 does not define the macro scale. It is actually a depiction at the micro scale or stand level. It would be extremely helpful for the authors to define what the macro scale is and to explain why one needs to study a phenomenon at this scale at all. I can imagine reasons, but the authors should have specific reasons in mind. Also it is not clear what the issues are. Finally, I don’t find these very compelling hypotheses. They just seem to justify the correlations that were examined, but don’t really examine any mechanisms.
Response:Our conceptual Figure 2 displays how climatic factors, soil factors, and stand factors together influence key leaf traits of forests. Regarding the macro scale you mentioned, we have already presented it very well in Figure 2.
115 I find it interesting that the trees do not seem to have any roots in the figure. I am not sure this figure does much more than just list factors, although the structural difference between planted and natural forests seems clearly illustrated.
Response: Our research did not focus on the underground parts of plants but concentrated on the traits of plant leaves. Therefore, the roots were not depicted.
119 Are there no methods? How is the reader supposed to understand the results if they have not been told the methods of analysis. Where did the data come from? How was it compiled or treated? How was it analyzed? Okay, I see that this information starts on line 298. It should go before the results.
Response: This is the basic format requirement of this journal.
122-123 This sentence does not make sense. The previous sentence compared two things. So is it the comparison that indicates a “rapid investment return strategy” or is it one of the two items being compared and which one?
Response: We have detailed in the Introduction: plants with a higher SLA and a lower LDMC adopt a rapid investment-return resource utilization strategy, while the opposite indicates a conservative strategy. Therefore, this strategy refers not just to a single indicator.
140 Is correlation needed? It really wasn’t a correlation analysis per se. I also do not understand how such a complex curve was determined from one variable. Why is there a periodicity to the model. What mechanisms supports this complexity in the models?
Response: This is the result of a generalized additive model, which is widely used. I don't quite understand why you would raise such a question; it is not a matter of so-called periodicity.
216 Is it the forests that adapt or is it the trees?
Response: Forests are composed of trees. The resource utilization strategy of a forest is actually a microcosm of plant resource utilization strategies.
229 I am not sure I am following the logic here. Natural forests are more diverse and include more species with differing traits which allows them to respond to a wider range of climatic conditions. But somehow this makes them less plastic than planted forests? There seems to be a disconnect. Is the issue that planted forests are more responsive to climatic changes? That is not clear from the text. Is more plastic the same as more sensitive?
Response: The reasons why natural forests exhibit lower plasticity in response to environmental changes compared to planted forests can be understood from several mechanistic perspectives: (1)Species in natural forests possess high genetic diversity, offering a broad adaptive capacity to environmental changes. This adaptability, based on species diversity, occurs through natural selection over extended periods, rather than rapid individual-level plastic changes. In contrast, species in planted forests are often selected for specific environmental conditions, with genetic variations more focused on rapid adaptation to these conditions, displaying higher individual plasticity. (2)Complex interactions among species in natural forests, including competition, symbiosis, and predation, create a stable ecological balance. This balance may render natural forests less swiftly adaptable to abrupt environmental changes in the short term compared to more simplified planted forests. Planted forests, typically consisting of a single or few tree species, reduce inter-species interactions, allowing for more direct and rapid responses to specific environmental changes.(3)Natural forests adapt to environmental changes through their complex structure and diverse functions, based on a variety of ecological processes and long-term evolution. Planted forests, with their simplified ecosystem structure and limited species composition, might adapt more quickly to environmental changes in the short term by increasing or decreasing specific functions. (4)Management of planted forests often aims to optimize certain economic or ecological services, such as timber production or rapid carbon sequestration, possibly through the selection of highly plastic species or the implementation of specific management practices. This deliberate intervention can enhance the rapid adaptability of planted forests to specific environmental changes. In contrast, management goals for natural forests are more varied and tend to focus on maintaining overall ecosystem health and diversity.In summary, the lower plasticity of natural forests compared to planted forests is mainly due to their reliance on species diversity and complex inter-species interactions within the ecosystem for adaptation to environmental changes, rather than rapid plastic responses of individual species. This system-based adaptation provides more enduring stability and resilience but may respond more slowly to rapid environmental changes.
281 Not according the title of the manuscript.
Response: We have changed the title as: Forest Age Drives the Resource Utilization Indicators of Trees in Planted and Natural Forests in China
384 I don’t think plasticity is the issue here; isn’t it responsiveness?
Response: We think plasticity is ok. Please see our paper: Gong et al., 2023; Gao et al., 2023.
387-389 Again this statement does not tell the reader anything new.
Response: We have excel this part.
300 I do not understand how forest coverage explains diversity or areal extent. This is not logical.
Response: The meaning of this passage is that China's forest coverage is very high, which facilitates our exploration of ecological patterns on a larger scale, not the issue you perceived.
314 What were these measurements applied to exactly? To leave collected from where and how?
Response: We have provided the message in the section of Methods.
how was abundance determined? The relative basal area? The number of stems? The mass of the leaves?
Response: Species abundance refers to the number of individuals of a species.
345 I am not sure what this means. What were the specific significance tests?
Response:We employed t tests at a 0.05 significance level to examine whether there are significant differences in SLA and LDMC between planted forests and natural forests.
Round 2
Reviewer 3 Report
Comments and Suggestions for Authors
While the authors responded adequately to some of the comments they tried to explain away others. My main issues, aside from the specific one indicated by lines below, are as follows:
The introduction is still not well organized. The first paragraph seems like a collection of unrelated items. Terms are not defined on their first use. Sentences the belong in one paragraph, based on the topic, are in another. There is a lack of topic sentences that would help the reader understand the flow of ideas.
I did not find Figure 1 very helpful in explaining what the macro scale was. It seems to suggest the macro scale is basically the stand level because that is what is depicted. But I get the sense by macro scale they are indicating a scale as extensive as China. Regardless the authors need to be clear and just because they know does not mean the reader knows. Given the number of times it is referred to the term just has to be defined. Ideally this would be done in the introduction.
There still are some issues regarding methods. Specifically, on exactly how the complex models in Figures 4 and 5 were created.
The authors need to discuss why models with an R2 of less than 10% are useful or provide insights. Almost any model can be significant at the 0.05 level if the degrees of freedom are large enough. That does not mean that it is important or useful in explaining biology or ecology.
It would be very helpful for the authors to discuss the application of these findings. How specifically would they help managers to manage forests better. What do they imply about response to climate change.
I encourage the authors to not explain away the concerns, but try to put themselves in the place of the reader that may not be an expert in their field. Also when a statement is ambiguous or could be taken the wrong way, they need to rephrase so that this is not an issue.
31 Can the authors be more specific? This is too generic a statement to have value.
41 The statement about forest area being planted is important. My question is why it is this paragraph which is largely about the two indicators. A paragraph should be about one set of ideas. This seems like a new idea. It would be far better, as I had suggested earlier, to more completely explain why understanding these strategies is important to know and how it might help in the management of forests.
43 The sentence starting on 41 belongs in this paragraph which seems to be about planted versus native forests. It just needs to be changed into a better topic sentence to start the paragraph.
45 nutrients or resources in general? The authors need to be consistent to avoid confusing the reader. Please check the entire manuscript.
48 would it not make more sense to define the terms in the first paragraph so that the reader does not have to wonder what the terms are exactly?
75 I suggest the authors start the sentence at soil nutrient factors. This is really the point of the sentence and it avoids the first part of the sentence which still does not entirely make sense.
107 I still do not see how Figure 1 helps the reader understand what the macro scale is.
117 the justification for not including the roots is that the roots were not studied. But the tree stems and branches were not studied and those seem to be on the figure. Clearly the N and P indicated in the soil would be absorbed and transported by roots. So I do not understand the reluctance to show the roots in the figure as the logic is inconsistent. It just seems to be the age old bias against displaying roots. The figure needs to be revised.
302 The authors did not address the confusion created by this sentence. I understand the authors know what they mean. The question is whether the reader will understand. It would be fine to state that forests cover 24% of China and hence are important. It is also fine to state that China’s forest occur in a wide range of conditions over a wide area, and reflect many types of forest biomes. But it is confusing to imply that 24% forest coverage explains any of those things. They could occur if the forest cover was 1% or 100%.
313 It is not clear what the consistent methods were. Many methods could be consistent, but it does not help the reader understand what the methods are.
317 Abundance is too vague a term. It could mean the number of stems per area, the basal area per area, the volume per area, or the biomass per area and in some cases a combination of these parameters. Please assist the reader in understanding what was used.
318 why is the word genuine needed? Is there such as thing as an artificial herbarium specimen?
366 It is still not clear how the very complex models in Figures 4 and 5 were created. I understand GAM was used. But the models are quite complex, at least more complex that a simple polynomial would suggest. Was some sort of smoothing function employed? It is important to fully describe how the data were treated so that the reader can understand what was done.
Author Response
Dear Reviewer,
I am the guest editor of this species issue and contribute this paper. We really greatly appreciate you for giving us an opportunity to revise our manuscript entitled “Forest Age Drives the Resource Utilization Indicators of Trees in Planted and Natural Forests in China” (Manuscript ID: plants-2867234) again. We are grateful for your valuable suggestions and comments on the manuscript. The point-by-point responses to your comments can be found below.
Yours sincerely,
Jie Gao and coauthors
########## Point-to-Point Response Letter ###############
The introduction is still not well organized. The first paragraph seems like a collection of unrelated items. Terms are not defined on their first use. Sentences the belong in one paragraph, based on the topic, are in another. There is a lack of topic sentences that would help the reader understand the flow of ideas.
Response: Thank you very much for your advice. We have revised the phrasing of the first paragraph. Key leaf traits refer to the morphological, physiological, and phenotypic characteristics of plants, which reflect their adaptability to the environment, survival strategies, and ecological niches [1]. Specific Leaf Area (SLA) and Leaf Dry Matter Content (LDMC) are commonly used to represent tree resource utilization strategies [1, 2]. SLA refers to the ratio of leaf area to its dry weight, while LDMC refers to the ratio of leaf dry weight to its fresh weight.Trees with higher SLA and lower LDMC are characterized by higher photosynthetic efficiency, minimized costs for leaf construction and maintenance, and maximized capacity for resource acquisition, indicative of a 'faster investment-return' resource utilization strategy [3]. Exploring the resource utilization strategies of forest trees holds significant value for the rational management and operation of forests.
I did not find Figure 1 very helpful in explaining what the macro scale was. It seems to suggest the macro scale is basically the stand level because that is what is depicted. But I get the sense by macro scale they are indicating a scale as extensive as China. Regardless the authors need to be clear and just because they know does not mean the reader knows. Given the number of times it is referred to the term just has to be defined. Ideally this would be done in the introduction.
Response: Thank you very much for your constructive suggestions. To make our research clearer to readers, we have removed the concept of macro-scale throughout the text.
There still are some issues regarding methods. Specifically, on exactly how the complex models in Figures 4 and 5 were created.
Response: Figures four and five present the results of the Generalized Additive Model. To clarify for the readers, we have listed the formula of the model. GAMs can be summarized as:
where g is a link function, E(Yi) is the estimate for the responsible variable Yi, S1 is the smooth function of xi over different light treatments, S2 is the smooth function of xi along investigation time, xi (i = 1, 2, 3,…, 12) are the explanatory variables, and they are number of new rhizomes, new rhizome length, new rhizome diameter, etc. β0 is constant term and ei is the error term.
The authors need to discuss why models with an R2 of less than 10% are useful or provide insights. Almost any model can be significant at the 0.05 level if the degrees of freedom are large enough. That does not mean that it is important or useful in explaining biology or ecology.
Response: In macro-scale studies, particularly global research, the R2 is generally very small. This is because different research sites exhibit strong habitat heterogeneity, and environmental factors hardly show linear changes along longitude, latitude, and altitude. As a result, the model's explanatory power is relatively smaller compared to local scales, but it still holds significant value.
It would be very helpful for the authors to discuss the application of these findings. How specifically would they help managers to manage forests better. What do they imply about response to climate change. I encourage the authors to not explain away the concerns, but try to put themselves in the place of the reader that may not be an expert in their field. Also when a statement is ambiguous or could be taken the wrong way, they need to rephrase so that this is not an issue.
Response: Exploring the resource utilization strategies of plantation and natural forest trees is of great importance for forest management and addressing global change. Firstly, by understanding the resource utilization strategies of different types of forests, managers can more effectively plan and implement forest management practices to enhance the growth efficiency of trees and the health of ecosystems. For example, understanding the photosynthetic efficiency and water use efficiency of specific tree species under certain environmental conditions can aid in selecting tree species that are more suited to local environments for planting and restoration efforts. Secondly, this knowledge is crucial for responding to global climate change. Different tree species respond differently to environmental factors such as temperature and precipitation, affecting their survival and reproduction under the backdrop of climate change. Understanding these resource utilization strategies helps predict the potential impacts of climate change on forest ecosystems and develop more targeted adaptation and mitigation measures. Moreover, comparing the resource utilization strategies of plantation and natural forests can reveal the impact of human intervention on the functionality of forest ecosystems, thus providing a scientific basis for ecological restoration and sustainable forest management. This includes optimizing the structure of tree plantations, enhancing the carbon sequestration capacity of forests, and increasing the resilience and adaptability of ecosystems to environmental changes.
31 Can the authors be more specific? This is too generic a statement to have value.
Response: Thanks for your suggestion. We have excel this sentence.
41 The statement about forest area being planted is important. My question is why it is this paragraph which is largely about the two indicators. A paragraph should be about one set of ideas. This seems like a new idea. It would be far better, as I had suggested earlier, to more completely explain why understanding these strategies is important to know and how it might help in the management of forests.
Response: Since natural forests are largely undisturbed by human activities, previous research on forests has mainly focused on natural forests [11,16]. However, from 1990 to 2020, the global area of planted forests increased by 123 million hectares [4–6]. Therefore, the ecological functions of planted forests, such as carbon sequestration, are gradually receiving attention. Compared to natural forests, planted forests typically have less species diversity, are more homogenous, and are largely subject to human management [6]. The differences, if any, in nutrient utilization strategies between trees in planted and natural forests, as well as the dominant factors and mechanisms due to different forest origins, remain unclear.
45 nutrients or resources in general? The authors need to be consistent to avoid confusing the reader. Please check the entire manuscript.
Response: Thanks for your idea. We have correct them.
48 would it not make more sense to define the terms in the first paragraph so that the reader does not have to wonder what the terms are exactly?
Response: Thanks for your suggestion. We have defined them in first paragraph.
75 I suggest the authors start the sentence at soil nutrient factors. This is really the point of the sentence and it avoids the first part of the sentence which still does not entirely make sense.
Response: Thanks for your idea. We have excel the first sentence.
107 I still do not see how Figure 1 helps the reader understand what the macro scale is.
Response: We have removed the concept of macro-scale throughout the text.
117 the justification for not including the roots is that the roots were not studied. But the tree stems and branches were not studied and those seem to be on the figure. Clearly the N and P indicated in the soil would be absorbed and transported by roots. So I do not understand the reluctance to show the roots in the figure as the logic is inconsistent. It just seems to be the age old bias against displaying roots. The figure needs to be revised.
Response: We have added the roots in the picture.
302 The authors did not address the confusion created by this sentence. I understand the authors know what they mean. The question is whether the reader will understand. It would be fine to state that forests cover 24% of China and hence are important. It is also fine to state that China’s forest occur in a wide range of conditions over a wide area, and reflect many types of forest biomes. But it is confusing to imply that 24% forest coverage explains any of those things. They could occur if the forest cover was 1% or 100%.
Response: Thanks for your suggestions. We have modified this part.
317 Abundance is too vague a term. It could mean the number of stems per area, the basal area per area, the volume per area, or the biomass per area and in some cases a combination of these parameters. Please assist the reader in understanding what was used.
Response: Thanks for your suggestions. We quantified the abundance (number of individuals) of each tree species within these plots.
318 why is the word genuine needed? Is there such as thing as an artificial herbarium specimen?
Response: The word “genuine” is removed.
366 It is still not clear how the very complex models in Figures 4 and 5 were created. I understand GAM was used. But the models are quite complex, at least more complex that a simple polynomial would suggest. Was some sort of smoothing function employed? It is important to fully describe how the data were treated so that the reader can understand what was done.
Response: Thanks for your suggestions. We have added the formula of GAMS. GAMs can be summarized as:
where g is a link function, E(Yi) is the estimate for the responsible variable Yi, S1 is the smooth function of xi over different light treatments, S2 is the smooth function of xi along investigation time, xi (i = 1, 2, 3,…, 12) are the explanatory variables, and they are number of new rhizomes, new rhizome length, new rhizome diameter, etc. β0 is constant term and ei is the error term.
Reviewer 1
I did not find the study very interesting, as we can find many parameters that will differ between natural forests and plantations, without having further impact on management. However, I have no formal objections, and therefore I will leave it to the editors to decide whether such a study is in the intention of the journal.
Response:Thank you very much for your advice. Comparing the functional traits related to resource utilization strategy of planted and natural forests holds significant value for forest management. In 2023, we published our findings on the critical role of functional traits in predicting the functions of planted and natural forests in the international journals Journal of Integrative Plant Biology and Forest Ecology and Management. Of course, we highly respect your evaluation and hope to gain your approval.
Reviewer 2
The manuscript, titled “Climatic Factors Determine the Resource Utilization Strategies 2 of Trees in Planted and Natural Forests at Macro Scales,” presents a study examining the influence of meteorological factors and soil on leaf size and dry matter content in natural and planted forests in China. The authors determined that planted forest trees exhibit higher ecological plasticity and greater variability depending on forest age in their resource use strategies than natural ones. The research is of interest for clarifying a number of environmental issues. However, there are a number of questions and comments regarding this study:
Response: Thank you for recognizing our work. We have responded to your comments one by one based on your suggestions.
- The final goal of this study is not entirely clear;
- Which strategy do the authors propose to use for natural forests, and which for planted forests?
- Have coniferous forests (natural and artificial) been studied and what dependencies were identified there?
- Was the height of forest growth and its influence on the factors studied taken into account?
- Was there any influence of zoning?
Response: Our study investigates the impact of climatic factors, soil factors, and stand factors on the nutrient utilization strategies of different forest types by comparing the Specific Leaf Area (SLA) and Leaf Dry Matter Content (LDMC) of planted and natural forests. This provides a theoretical basis for understanding the effects of global change on forest management. Both resource utilization strategies identified in this study are applicable to planted and natural forests. However, the nutrient utilization strategies of these two types of forests may change in response to variations in climatic factors, soil nutrients, and stand factors. Coniferous forests exist in both planted and natural forests; however, this study focused solely on planted and natural forests. Thank you very much for your feedback; it will be a key focus of our future research. Generally speaking, forest height is closely related to forest age and stand density. Older forests typically have taller trees. Zoning may have an influence, but different zones have unique climate and soil conditions. Therefore, when exploring the specific impacts of climate and soil factors, we also quantified the contribution of zoning to a certain extent.
Reviewer 3
Thank you very much for the valuable comments you provided on this article. I am the column editor for the journal Plants, where I have established a column on plant responses to climate change and contributed this article. My main research areas are forestry and ecology. I have carefully revised the article in response to your comments, hoping for mutual learning. My specific responses to your review comments are as follows:
General comments:
While the manuscript examines a topic that should be of considerable interest, it is need of significant improvement. This includes:Moving the methods to before the results so that the reader can understand how the analysis was conducted.
Response: Thank you for your suggestion; however, this is the format required by MDPI, and I hope you can understand.
More details on methods in terms of where leaf samples were collected, how they were collected and treated, how literature values were integrated with field data.
Response: SLA (Specific Leaf Area) and LDMC (Leaf Dry Matter Content) are widely used to indicate plant resource utilization strategies [3,46,47]. In each designated forest plot, we meticulously collected leaves that were free from any pest or disease damage from more than 20 mature and fully grown trees per species. These samples were carefully harvested from the tree crowns' outer branches using scissors, ensuring minimal disturbance to the tree structure. Subsequently, the leaves were sandwiched between two layers of damp filter paper and securely stored in zip-lock bags for transport to the lab. Upon arrival, these samples were promptly placed in a refrigeration unit. Before conducting further analysis, we removed the leaves from their bags and allowed them to incubate for 12 hours in a dark environment at 5℃ to rid them of any superficial moisture. Finally, the leaves were delicately patted dry using filter paper to absorb any residual water before their saturated fresh weight was accurately measured on an electronic scale.
In our experiments, we utilized the Japanese Cano Scan LIDE 110 portable leaf area meter to measure the area of freshly picked single leaves with petioles removed. The fresh weight of the leaves was determined using an electronic scale with a precision of 0.0001g. Fresh leaves were then placed in an oven at 105°C for 15 minutes to kill green tissue, followed by drying at 60°C for 48 hours, after which their dry weight was measured using an electronic scale accurate to one ten-thousandth. The Specific Leaf Area SLA (m² / kg) is calculated as leaf area divided by leaf dry weight, and Leaf Dry Matter Content LDMC (g / g) as leaf dry weight divided by leaf fresh weight.
In this study, the dataset encompassing 697 forests from 82 locations comprised two primary sources: 438 forest datasets from 51 sites were gathered through literature review, while the remaining 259 forest datasets from 31 sites were obtained from experiments conducted in this research. The data extracted from the literature were processed and calculated according to consistent experimental methods.
The findings are presented as definitive and causal, but are far from that. Correlational analyses were conducted and those cannot prove a causal relationship. They can suggest them, but that is not a definitive result. Also some of the correlations are extremely weak, yet are presented as if they all have biological significance. The authors need to be far more careful about how their results are presented and qualified.
Response: Thank you very much for your advice. In ecology or botany, there is never a direct cause-and-effect relationship. All relationships are presented based on correlation coefficients or the goodness of fit of models. We fully understand your concerns, but I believe we can discuss some practical issues through correlation coefficients, R2, or p-values. Frankly speaking, we have published several papers in CNS sub-journals (Nature climate change or Current biology), where, in macro studies, R2 might not exceed 0.1, but it still holds statistical value.Additionally, we employed structural equation modeling, a model widely used for the indirect relationships among multivariate. In any case, we are very grateful for your advice and will pay extra attention to this aspect in our subsequent research.
It is never clear what a macro scale is. It seems to be a key justification for the study, yet it is never really defined or the issues with micro scale analyses being solved made clear. From what I can tell they did in fact do some analysis at the micro scale and so it is very confusing as to what they are learning really is.
Response: Thanks for your views. We have clearly presented our research scale in Figure 2. The latitude range of China is 49 degrees, and the longitude range is 62 degrees. Therefore, if this scale does not qualify as a macro scale, it's hard for me to imagine that only a global study would. Whether it's considered a macro scale is not the focus of our research; we simply want readers to understand the biological patterns presented at such a broad geographical scale. Of course, to respect your opinion, we have removed the concept of "macro scale" from the title of our paper, hoping you can understand.
There are contradictions in the manuscript that do not seem to make any sense. Some of this may be related to an imprecision of terminology. For example, the authors refer to plasticity, but what they are really discussing is responsiveness.
Response: Plasticity refers to the degree of variation in the dependent variable as environmental factors change. Generally, we represent plasticity with the model's R2. This approach is widely used in the journal of Science or Nature.
2 Given this was a correlational study I do not think the word “determine” is appropriate. That implies a cause and effect relationship that cannot be determined using correlation. That is an overstatement of what the methods can tell one.
Response: I agree with your perspective. Therefore, we have changed "determine" to "drive." We hope this phrasing better aligns with your thoughts.
3 The location of the study should be made clear in the title.
Response: Thanks for your suggestions. We have added the location in the title.
14 “Due to differences in forest origins” is not needed as this is covered by latter part of the sentence.
Response: Thanks for your suggestions. We have excel this statement.
18 There is nothing about the methods of how the analysis was done. These should be mentioned so that the reader understands the results summary.
Response:This study, based on field surveys and literature from 2008 to 2020 covering 263 planted and 434 natural forests in China, using generalized additive models (GAMs) and structural equation models (SEMs) analyze the spatial differences and dominant factors in tree resource utilization strategies between planted and natural forests.
22 These seems to be no uncertainty in the results. I doubt this was the case. I suggest the authors qualify these statements to be less definitive.
Response:Thank you for your suggestion. Our statements are based on our research findings, and our results do not contain absolute terms (such as "definitely," "completely," etc.).
24-25 These statements seem to contradict the title. Which is correct?
Response: We have changed our title as “Forest Age Drives the Resource Utilization Indicators of Trees in Planted and Natural Forests in China”.
28-30 This concluding sentence does not tell the reader anything new. It also employs circular logic. The study was done on X , so now we know more about X. Are there specific implications of the findings?
Response: We have changed the statement: These findings have significant ecological implications for forest management.
35 Here nutrients are mentioned, but on line 11 it was resources. Which is it? Please be consistent. Also please define the terms themselves so that the reader can understand the results and their implications more easily.
Response: Thanks for your suggestions. We have changed the statement. Key leaf traits directly reflect tree resource utilization strategies.
41 This sentence starts a completely new topic. It should start a new paragraph.
Response: We agree with your opinions. This sentence starts a new paragraph.
42 Is the point that the planted forests are more directly impacted? Both are impacted by human management.
Response: The reasons why planted forests are more susceptible to impacts are: (1) Planted forests are usually composed of a single or a few tree species, lacking the biodiversity found in natural forests. This monoculture makes planted forests more vulnerable to specific pests and diseases due to the absence of natural control mechanisms and inter-species support. (2) Natural forests have developed a relatively stable ecological balance through long periods of evolution, whereas planted forests, with their shorter history of plantation and simpler ecosystems, lack sufficient time and conditions to form a stable ecological equilibrium, making them more susceptible to external disturbances. (3) The choice of tree species and growth conditions in planted forests are often based on economic or specific utility considerations rather than natural selection within ecosystems. This may lead to trees that are not adapted to local environmental conditions, thereby making them more susceptible to drought, diseases, and other impacts.
44 What is the macro scale? Please define. Also it is not clear why this would be of interest.
Response: In plant ecology, a very popular topic is the macro-scale changes characteristic of plant ecology. Common scales of study include macro scale, regional scale, and individual scale. This study is based in China, and therefore, it is on a macro scale.
35-45 I believe the problem is that this section does not do a very good job at introducing the manuscript. It seems to generally cover ideas that are more fully covered below. Perhaps this paragraph should be about the importance of understanding tree nutrient/resource strategies?
Response: Thank you for your feedback. This section is crucial for informing readers about the importance of tree survival strategies in responding to global changes. This is very important for those interested in studying plant ecology at a macro scale.
59 increase specific leaf area or leaf area?
Response: Thanks for your suggestions. That should be :increase specific leaf area.
68-69 Some of the logic behind this conclusion seems to be missing. Is this true because species have different strategies? Or because management creates different resource availabilities?
Response: Thanks for your advice. The differences in species composition and management between planted and natural forests imply significant variations in how different species respond to climate change in terms of resource utilization strategies.
73 Don’t parts of the tree live outside the soil? This seems to suggest otherwise.
Response: Thank you very much for your suggestion. From the perspective of plant ecology, it is true that parts of plants live in the air. However, plants primarily rely on the soil for their nutrient and water uptake. We have added the word "mainly" to clarify this point.
96 please define stand density. Do the authors mean the number of trees per area or something else here? Also I find it hard to believe that the light resource is more abundant. Isn’t that determined by the length of day, the day of the year, the latitude, and the weather? How does stand density influence any of that? Do the authors mean that more of the light resource is available?
Response: Stand density typically refers to the number of trees within a given area, which is one of the key indicators for measuring the structure of a forest or stand. Indeed, the total amount of light resources is determined by factors such as the length of day, the date within the year, latitude, and weather conditions, which are not affected by stand density. However, stand density affects the distribution and efficiency of light resource utilization beneath the canopy. In denser stands, competition among trees, especially for light, is more intense, which may result in reduced light reaching the understory and forest floor. Therefore, when mentioning that light resources are "more abundant," it could mean that in stands with lower density, the ground layer and understory plants have the opportunity to receive more light, thereby increasing the overall light utilization efficiency of the stand.
99 Is it solely a function of complexity? Isn’t largely due to the fact the canopy is more developed?
Response: In the later stages of stand development, the complexity of forest functions is not solely due to the development of the canopy. Although the development of the canopy directly impacts the forest's microclimate, light distribution, water cycle, and the growth of understory plants, thereby determining the ecological functions of the forest to some extent, the complexity of forest functions is influenced by a variety of factors. Different tree species and types of plants have varying impacts on ecosystem functions, and interactions between species also add to the system's complexity. The type, structure, and nutrient content of the soil have significant effects on the forest's water retention capability and nutrient cycling. Besides the canopy layer, the vertical and horizontal structure of the stand, including the understory vegetation and litter layer, also impacts the ecological functions of the forest.
105 unfortunately figure 1 does not define the macro scale. It is actually a depiction at the micro scale or stand level. It would be extremely helpful for the authors to define what the macro scale is and to explain why one needs to study a phenomenon at this scale at all. I can imagine reasons, but the authors should have specific reasons in mind. Also it is not clear what the issues are. Finally, I don’t find these very compelling hypotheses. They just seem to justify the correlations that were examined, but don’t really examine any mechanisms.
Response:Our conceptual Figure 2 displays how climatic factors, soil factors, and stand factors together influence key leaf traits of forests. Regarding the macro scale you mentioned, we have already presented it very well in Figure 2.
115 I find it interesting that the trees do not seem to have any roots in the figure. I am not sure this figure does much more than just list factors, although the structural difference between planted and natural forests seems clearly illustrated.
Response: Our research did not focus on the underground parts of plants but concentrated on the traits of plant leaves. Therefore, the roots were not depicted.
119 Are there no methods? How is the reader supposed to understand the results if they have not been told the methods of analysis. Where did the data come from? How was it compiled or treated? How was it analyzed? Okay, I see that this information starts on line 298. It should go before the results.
Response: This is the basic format requirement of this journal.
122-123 This sentence does not make sense. The previous sentence compared two things. So is it the comparison that indicates a “rapid investment return strategy” or is it one of the two items being compared and which one?
Response: We have detailed in the Introduction: plants with a higher SLA and a lower LDMC adopt a rapid investment-return resource utilization strategy, while the opposite indicates a conservative strategy. Therefore, this strategy refers not just to a single indicator.
140 Is correlation needed? It really wasn’t a correlation analysis per se. I also do not understand how such a complex curve was determined from one variable. Why is there a periodicity to the model. What mechanisms supports this complexity in the models?
Response: This is the result of a generalized additive model, which is widely used. I don't quite understand why you would raise such a question; it is not a matter of so-called periodicity.
216 Is it the forests that adapt or is it the trees?
Response: Forests are composed of trees. The resource utilization strategy of a forest is actually a microcosm of plant resource utilization strategies.
229 I am not sure I am following the logic here. Natural forests are more diverse and include more species with differing traits which allows them to respond to a wider range of climatic conditions. But somehow this makes them less plastic than planted forests? There seems to be a disconnect. Is the issue that planted forests are more responsive to climatic changes? That is not clear from the text. Is more plastic the same as more sensitive?
Response: The reasons why natural forests exhibit lower plasticity in response to environmental changes compared to planted forests can be understood from several mechanistic perspectives: (1)Species in natural forests possess high genetic diversity, offering a broad adaptive capacity to environmental changes. This adaptability, based on species diversity, occurs through natural selection over extended periods, rather than rapid individual-level plastic changes. In contrast, species in planted forests are often selected for specific environmental conditions, with genetic variations more focused on rapid adaptation to these conditions, displaying higher individual plasticity. (2)Complex interactions among species in natural forests, including competition, symbiosis, and predation, create a stable ecological balance. This balance may render natural forests less swiftly adaptable to abrupt environmental changes in the short term compared to more simplified planted forests. Planted forests, typically consisting of a single or few tree species, reduce inter-species interactions, allowing for more direct and rapid responses to specific environmental changes.(3)Natural forests adapt to environmental changes through their complex structure and diverse functions, based on a variety of ecological processes and long-term evolution. Planted forests, with their simplified ecosystem structure and limited species composition, might adapt more quickly to environmental changes in the short term by increasing or decreasing specific functions. (4)Management of planted forests often aims to optimize certain economic or ecological services, such as timber production or rapid carbon sequestration, possibly through the selection of highly plastic species or the implementation of specific management practices. This deliberate intervention can enhance the rapid adaptability of planted forests to specific environmental changes. In contrast, management goals for natural forests are more varied and tend to focus on maintaining overall ecosystem health and diversity.In summary, the lower plasticity of natural forests compared to planted forests is mainly due to their reliance on species diversity and complex inter-species interactions within the ecosystem for adaptation to environmental changes, rather than rapid plastic responses of individual species. This system-based adaptation provides more enduring stability and resilience but may respond more slowly to rapid environmental changes.
281 Not according the title of the manuscript.
Response: We have changed the title as: Forest Age Drives the Resource Utilization Indicators of Trees in Planted and Natural Forests in China
384 I don’t think plasticity is the issue here; isn’t it responsiveness?
Response: We think plasticity is ok. Please see our paper: Gong et al., 2023; Gao et al., 2023.
387-389 Again this statement does not tell the reader anything new.
Response: We have excel this part.
300 I do not understand how forest coverage explains diversity or areal extent. This is not logical.
Response: The meaning of this passage is that China's forest coverage is very high, which facilitates our exploration of ecological patterns on a larger scale, not the issue you perceived.
314 What were these measurements applied to exactly? To leave collected from where and how?
Response: We have provided the message in the section of Methods.
how was abundance determined? The relative basal area? The number of stems? The mass of the leaves?
Response: Species abundance refers to the number of individuals of a species.
345 I am not sure what this means. What were the specific significance tests?
Response:We employed t tests at a 0.05 significance level to examine whether there are significant differences in SLA and LDMC between planted forests and natural forests.